# The clinical significance of stringent complete response in multiple myeloma is surpassed by minimal residual disease measurements

**Maria-Teresa Cedena**[1,2], **Estela Martin-Clavero**[1,2], **Sandy Wong**[3], **Nina Shah**[3], **Natasha Bahri**[3], **Rafael Alonso**[1,2], **Carmen Barcenas**[4], **Antonio Valeri**[1,2], **Johny Salazar Tabares**[4], **Jose Sanchez-Pina**[1,2], **Clara Cuellar**[1,2], **Thomas Martin**[3], **Jeffrey Wolf**[3], **Juan-Jose Lahuerta**[1,2☉], **Joaquin Martinez-Lopez**[1,2,3☉]*

1 Hematology Department, Hospital Universitario 12 de Octubre, Madrid, Spain, 2 Instituto de Investigación Sanitaria imas12, H12O-CNIO Hematological Malignancies Research Group, CIBERONC, Complutense University, Madrid, Spain, 3 Bone Marrow Transplantation and Hematologic Malignancy Unit, Division of Hematology-Oncology, University of California, San Francisco, CA, United States of America, 4 Pathology Department, Hospital Universitario 12 de Octubre, Madrid, Spain

☉ These authors contributed equally to this work.
* jmarti01@ucm.es, jmartinezlo@yahoo.es

**Data Availability Statement:** All relevant data are within the paper.

## Abstract

### Background

Stringent complete response (sCR) is used as a deeper response category than complete response (CR) in multiple myeloma (MM) but may be of limited value in the era of minimal residual disease (MRD) testing.

### Methods

Here, we used 4-colour multiparametric flow cytometry (MFC) or next-generation sequencing (NGS) of immunoglobulin genes to analyse and compare the prognostic impact of sCR and MRD monitoring. We included 193 treated patients in two institutions achieving CR, for which both bone marrow aspirates and biopsies were available.

### Results

We found that neither the serum free light chain ratio, clonality by immunohistochemistry (IHC) nor plasma cell bone marrow infiltration identified CR patients at distinct risk. Patients with sCR had slightly longer progression-free survival. Nevertheless, persistent clonal bone marrow disease was detectable using MFC or NGS and was associated with significantly inferior outcomes compared with MRD-negative cases.

### Conclusion

Our results confirm that sCR does not predict a different outcome and indicate that more sensitive techniques are able to identify patients with differing prognoses. We suggest that MRD categories should be implemented over sCR for the future classification of MM responses.

**Funding:** This study was supported by grants of the Instituto de Salud Carlos III FEDER founds (Ministry of Economy and Competitivity, Madrid, Spain) FIS PI15/01484 and CRIS foundation grants 14/001 to JML. The funders had no role in study design, data collection and analysis, decision to publish, or preparation of the manuscript.

**Competing interests:** J Martinez-Lopez belongs to the speaker bureau of Adaptive Biotechnologies. This does not alter our adherence to PLOS ONE policies on sharing data and materials. The rest of the authors declare no competing financial interests.

## Introduction

The precise classification of the response to treatment in multiple myeloma (MM) is crucial for the evaluation of clinical efficacy and prognosis [1, 2], and improving the definition of response is becoming increasingly relevant. Four categories of deep response have been established by the International Myeloma Working Group (IMWG) [3] based on various criteria and methodologies, but in our opinion, some of these categories are redundant.

The IMWG introduced the stringent complete response (sCR) criteria in 2006 [4] by adding a criterion for the normalization of the kappa and lambda serum free light chain (sFLC) ratio plus the absence of clonal plasma cells in bone marrow (BM) by immunohistochemistry (IHC) or immunofluorescence with a sensitivity of $10^{-3}$ [5, 6]. The sCR criteria have remained with minor changes in subsequent reviews [3, 7]. However, the value of the sCR criteria over CR could be controversial.

Only one study reported the impact of achieving sCR after autologous stem cell transplantation in MM patients [8], while other authors did not find that the normalization of the κ/λ ratio provides more additional prognostic value [9–11].

In a previous work from our group, we showed that the sCR criteria only identified a small group of patients with dismal prognosis, namely, those with high residual burden by low-sensitivity MFC [12]. We also found that sFLC ratio normalization in patients in CR did not identify patients with differing outcomes [12, 13].

With the emergence of new and more effective treatments along with sensitive techniques to study minimal residual disease (MRD), the role of sCR is probably becoming less relevant. Additionally, the assessment of clonality by IHC requires the performance of a BM biopsy, which is more painful and expensive than aspiration.

Here, we analysed and compared the prognostic value of sCR and MRD monitoring by intermediate-sensitivity flow cytometry (first- and second-generation) or next-generation sequencing of the immunoglobulin genes in patients with conventional CR treated in two institutions.

## Patients and methods

### Patients and samples

The study group included patients with MM in CR treated at Hospital 12 Octubre, Madrid or the University of California, San Francisco between 2003 and 2018. The patients were treated according to usual clinical protocols, and bone marrow (BM) samples were obtained to assess the response of therapy following the usual practice in each centre. This retrospective observational study was approved by the ethics committees of Hospital 12 Octubre and University of California San Francisco (UCSF), and all patients provided written informed consent for the use of the samples. All procedures performed were in accordance with the ethical standards of institutional and/or national research committees and with the 1964 Helsinki Declaration and its later amendments or comparable ethical standards.

### Methods

BM clonality was defined by IHC when the κ/λ ratio was >4:1 or <1:2 for κ and λ patients, respectively. sFLC was measured by immune-nephelometry (FREELITE assay; Binding Site Ltd., Birmingham, UK), and sFLC κ/λ ratios were classified as normal (0.26–1.65) or abnormal (<0.26 if the patient was λ; >1.65 if the patient was κ) following IMWG guidelines [7]. Plasma cell infiltration in BM aspirates was also evaluated in May-Grünwald-Giemsa-stained smears by conventional cytomorphology, counting 500 nucleated cells.

BM biopsies were obtained from the iliac crest with an 11G trephine biopsy needle and were fixed in 5% formalin. The samples were decalcified and embedded in paraffin, and 2-μm sections were stained with haematoxylin and eosin. BM biopsy IHC was carried out on paraffin sections as previously described [14]. BM plasma cell percentages were calculated by counting 500 nucleated cells on BM smears by two different pathologists.

MRD by MFC was performed using conventional 4-colour MFC as previously described [15, 16]. In brief, erythrocyte-lysed whole-BM samples were immunophenotyped using the 4-colour antibody combination CD38-FITC/CD56-PE/CD19-PerCP-Cy5.5/CD45-APC. The data acquisition of 2 million cells was performed with FACSCalibur and FACSCanto II flow cytometers (Becton-Dickinson, San Jose, CA); specific software was used to analyse the flow cytometry data [17], and the expected sensitivity was $10^{-4}$–$10^{-5}$.

MRD by NGS of immunoglobulin genes was performed by commercially available NGS (Adaptive Biotechnologies, Seattle, WA), as previously described [18]. Patients in whom a high frequency of myeloma clones (>5%) were not identified were excluded from the MRD analysis. MRD was assessed in patients with a high frequency of myeloma clones using the IGH-VDJH and IGK or IGH-VDJH, IGH-DJH, IGK and IGL assays. Once the absolute amount of total cancer-derived molecules present in a sample was determined, a final MRD measurement was calculated, providing the number of cancer-derived molecules per 1 million cell equivalents.

## Statistical analysis

All data were stored in a REDCap database (Vanderbilt University, Nashville, TE) and in MS Excel files. Statistical analysis was performed with SPSS version 22.0 (Statistical Package for Social Sciences Inc., Chicago, IL). Progression-free survival (PFS) curves were plotted by the Kaplan-Meier method, and the log-rank test was used to estimate the statistical significance of differences in survival rates. $P$-values <0.05 were considered significant. The χ2 and Fisher's exact two-sided tests were used for comparisons between categorical variables, and the Wilcoxon rank sum test or t-test was used for continuous variables. Correlation and concordance analyses were performed with Pearson R and Kappa index tests.

## Results

The study comprised 193 patients treated at Hospital 12 Octubre (n = 61) or the University of California, San Francisco (n = 132). Eighty percent of patients were transplant-eligible, and a similar number received some type of maintenance, mainly lenalidomide. The median follow-up was 27 months. All patients had CR. The primary clinical characteristics are summarized in Table 1.

A total of 161 BM samples from 130 patients were analysed with both IHC and MFC, whereas 98 BM samples were analysed with both IHC and NGS of immunoglobulin gen.

## Comparison of immunohistochemistry with conventional cytomorphology

In a subset of 92 samples, we used three methods to identify patients with persistent MM disease: 9/92 samples (10%) were considered clonal by IHC, 11 samples (11%) had more than 5% plasma cells by conventional cytology, and 38/92 (41%) were considered positive by MFC. We found low correlations between plasma cell infiltration evaluated by cytological techniques and IHC or MFC (R = 0.27 and 0.56, respectively) and between that evaluated by IHC and MFC (R = 0.41).

**Table 1. Main clinical characteristics of the included patients.**

| Characteristic | | n (193) |
|---|---|---|
| Male/female, % | | 57/43 |
| Mean age, years (range) | | 58.8 (28–82) |
| Mean haemoglobin, mg/dL (SD) | | 10.6 (2.2) |
| Mean creatinine, mg/dL (SD) | | 1 (1.1) |
| LDH high, % | | 21 |
| High-risk cytogenetics†, % | | 19 |
| International Staging System, (%) | | |
| | I | 36 |
| | II | 29 |
| | III | 35 |
| Myeloma type | | |
| | IgG | 55 |
| | IgA | 20 |
| | LC | 20 |
| | Other | 5 |
| Treatment, (%) | | |
| | VRD or VTD | 48 |
| | KRD | 3 |
| | RD | 4 |
| | Keyboard | 21 |
| | Others | 24 |
| Autologous Stem Cell Transplantation (Y/N), % | | 80/20 |
| Maintenance (Y/N), % | | 82/18 |

†del(17p) and/or translocation t(4;14) and/or translocation t(14;16).

Abbreviations: SD: standard deviation; LC: light chain; VRD: bortezomib-lenalidomide-dexamethasone; VTD: bortezomib-thalidomide-dexamethasone; KRD: carfilzomib-lenalidomide-dexamethasone; RD: lenalidomide-dexamethasone; CyBorD: cyclophosphamide-bortezomib-dexamethasone.

## Prognostic impact of stringent complete response, immunohistochemistry and serum free light chain

Patients not showing clonality (n = 162) by IHC and those showing clonality (n = 31) had similar PFS outcomes (106 vs 122 months, 95% confidence interval [95% CI]: ND and 66–140, respectively, $P = 0.5$, Fig 1A), and patients with less than 5% of plasma cells by cytology had

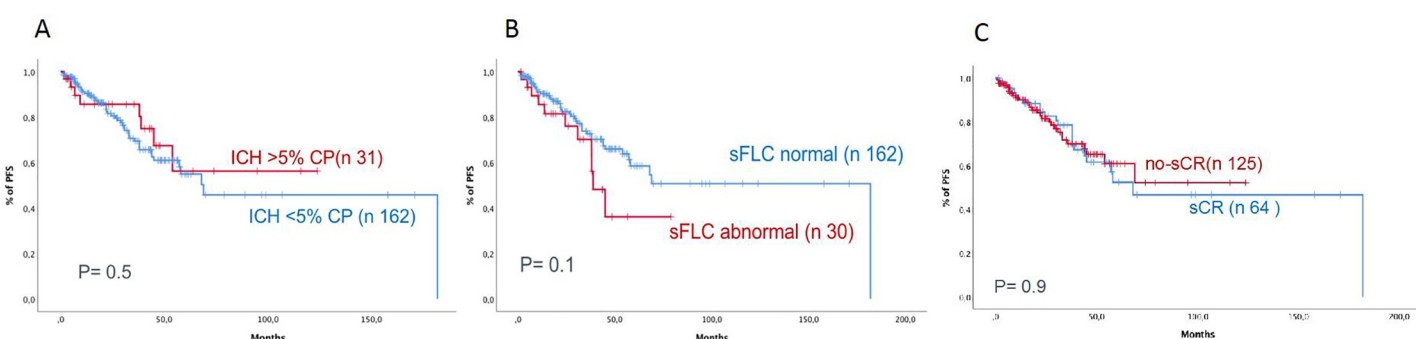

**Fig 1.** Progression-free survival of patients based on (A) immunohistochemistry, (B) normal serum free light chain ratio, and (C) stringent complete response.

the same PFS as those with more than 5% (89 vs 114 months, 95% CI: 11–89 and 87–142 months, respectively, $P$ = 0.6).

Patients with normalized sFLC ratios (n = 162) and patients with abnormal sFLC ratios (n = 30) had similar outcomes (PFS 38 months vs ND, 95% CI: NA and 38–41 months, respectively, $P$ = 0.1, Fig 1B).

Next, we investigated whether patients with sCR (n = 64) and patients not achieving sCR (n = 125) had similar outcomes and found that both groups had similar outcomes (PFS 68 vs 69 months, 95% CI: 27–108 months and ND, respectively, $P$ = 0.5, Fig 1C).

## MRD assessment

Patients with negative MFC results had superior outcomes to patients failing to reach MRD negativity (PFS NA vs 53 months, 95% CI: NA and 32–77 months, respectively, $P$ = 0.02). We then explored whether MFC identified patients with dismal outcomes among patients with sCR. A total of 108 patients with sCR were MRD-negative by MFC and had better outcomes than MRD-positive patients (n = 22) (PFS 58 months vs NA, 95% CI 35–71 months and ND, respectively, $P$ = 0.04, Fig 2A).

Finally, patients MRD-negative by NGS had superior outcomes to patients failing to reach MRD negativity (PFS NA vs 38 months, 95% CI: NA and 27–49 months, respectively, $P$ = 0.0001). We explored whether NGS identified patients with dismal outcomes among patients with sCR. A total of 34 patients with sCR were MRD-negative by NGS and had better outcomes than MRD-positive patients (n = 64) (PFS 32 months vs. ND, 95% CI: 22–42 months and ND, respectively, $P$ = 0.001, Fig 2B).

## Discussion

To our knowledge, this is the first study that performed a head-to-head comparison of sCR and MRD to define patients with different outcomes. sCR has been hypothesized to represent a deep response; however, both sFLC and plasma cell clonality in BM have limitations in defining patients with dismal outcomes in an era of highly sensitive techniques, including MFC and molecular approaches.

Despite the limitations of our study (retrospective nature, relatively small sample of patients, low sensitivity of MFC), we found that neither BM clonality by IHC nor plasma cell

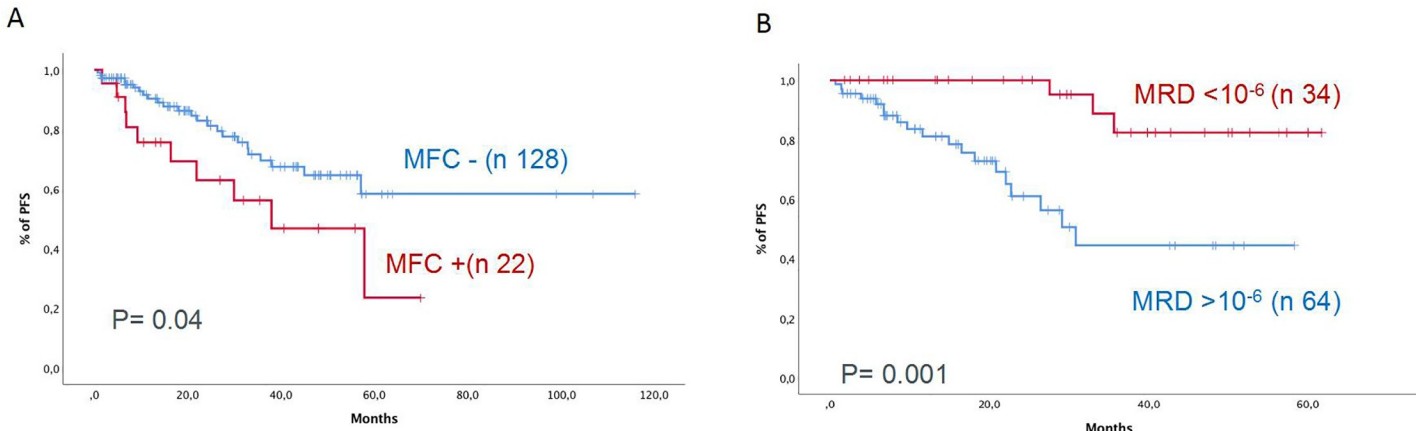

**Fig 2.** Progression-free survival of patients with stringent complete response classified by multiparametric flow cytometry (MFC) (A) and next-generation sequencing (NGS) results (B).

counts could identify patients with better outcomes, which contrasts with other studies showing the clinical impact of both methodologies. Low sensitivity and lack of specificity could explain this difference, but the efficacy of new maintenance treatments could also influence these results; approximately 80% of patients were treated with maintenance therapy. It is well recognized that BM biopsy has more adverse events and is more painful than BM aspiration, so avoiding BM biopsy in the response evaluation of MM is of value to MM patients.

The sFLC test to identify the monoclonal or polyclonal nature of the immunoglobulin light chains and an altered κ/λ ratio by oligoclonal bands [19] has emerged in the context of immune regeneration [20], and we report a lack of clinical relevance for κ/λ ratios, similar to previous observations. The absence of significant differences in progression-free survival and overall survival between patients in the stringent and conventional CR groups differs from that reported by Kapoor *et al.* [10], and these findings reinforce the fact that sFLC does not contribute additional clinical information to immunofixation to define conventional CR in MM. This affirmation does not mean that sFLC is not an adequate method to follow MM patients; sFLC might be considered as an alternative method to define CR, but it is not superior. Future investigations using more sensitive methods, such as mass spectrophotometry, to identify monoclonal proteins in serum are needed [21].

MRD assessment is currently being considered a new therapeutic objective in MM. MFC is a sensitive technique that can capture these differences and identify patients with different prognoses [22, 23]. We previously showed that MRD monitoring by conventional low-sensitivity MFC, but not defined by IHC, has a similar outcome prediction to IHC but does not have any clinical consequences; however, intermediate-sensitivity MFC could improve clinical prediction [24]. These results underscore and confirm the fact that attaining deeper levels of remission translates into prolonged PFS. The MRD methodology described in this study, although using 4-colour MFC, has an intermediate sensitivity above $10^{-4}$ due to the acquisition of more events (approximately 2 million). This sensitivity is much higher than that achieved by IHC, which is approximately 1%. Further, the predictions of outcome could be improved in the future with next-generation flow cytometry [25]. In fact, a recently published study confirmed that MRD-negative patients, determined by using next-generation flow cytometry methodology, had an 82% reduction in the risk of progression or death [26].

Finally, NGS of immunoglobulin genes is a step forward in the definition of deep response in MM [24, 27, 28]. We showed that MRD by NGS could differentiate patients with different prognoses achieving sCR or CR with high accuracy.

In summary, our results confirm that response assessment according to the stringent CR criteria does not predict a different outcome for MM patients with CR. However, more sensitive techniques, including both MFC and NGS of immunoglobulin genes, might identify patients with different prognoses, even among patients with sCR. These results confirm our previous findings and strengthen our suggestion that the sCR category should be rethought for the future classification of MM response.

## Author Contributions

**Conceptualization:** Maria-Teresa Cedena, Juan-Jose Lahuerta, Joaquin Martinez-Lopez.

**Data curation:** Estela Martin-Clavero, Sandy Wong, Nina Shah, Natasha Bahri, Rafael Alonso, Carmen Barcenas, Jose Sanchez-Pina, Clara Cuellar, Thomas Martin, Jeffrey Wolf, Juan-Jose Lahuerta, Joaquin Martinez-Lopez.

**Formal analysis:** Maria-Teresa Cedena, Estela Martin-Clavero, Sandy Wong, Nina Shah, Natasha Bahri, Rafael Alonso, Carmen Barcenas, Antonio Valeri, Johny Salazar Tabares,

Jose Sanchez-Pina, Clara Cuellar, Thomas Martin, Jeffrey Wolf, Juan-Jose Lahuerta, Joaquin Martinez-Lopez.

**Funding acquisition:** Jeffrey Wolf, Juan-Jose Lahuerta, Joaquin Martinez-Lopez.

**Investigation:** Estela Martin-Clavero, Nina Shah, Natasha Bahri, Rafael Alonso, Carmen Barcenas, Antonio Valeri, Jose Sanchez-Pina, Clara Cuellar, Thomas Martin, Juan-Jose Lahuerta, Joaquin Martinez-Lopez.

**Methodology:** Maria-Teresa Cedena, Estela Martin-Clavero, Carmen Barcenas, Johny Salazar Tabares, Jeffrey Wolf, Juan-Jose Lahuerta, Joaquin Martinez-Lopez.

**Project administration:** Joaquin Martinez-Lopez.

**Resources:** Maria-Teresa Cedena, Estela Martin-Clavero, Sandy Wong, Juan-Jose Lahuerta, Joaquin Martinez-Lopez.

**Supervision:** Juan-Jose Lahuerta, Joaquin Martinez-Lopez.

**Validation:** Juan-Jose Lahuerta, Joaquin Martinez-Lopez.

**Visualization:** Juan-Jose Lahuerta, Joaquin Martinez-Lopez.

**Writing – original draft:** Maria-Teresa Cedena, Juan-Jose Lahuerta, Joaquin Martinez-Lopez.

**Writing – review & editing:** Maria-Teresa Cedena, Estela Martin-Clavero, Sandy Wong, Nina Shah, Natasha Bahri, Rafael Alonso, Carmen Barcenas, Antonio Valeri, Johny Salazar Tabares, Jose Sanchez-Pina, Clara Cuellar, Thomas Martin, Jeffrey Wolf, Juan-Jose Lahuerta, Joaquin Martinez-Lopez.

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
