## [Decision Letter · Decision Letter 0]

10 Jun 2020

PONE-D-20-12394

Clinical significance of stringent complete response in multiple myeloma is surpassed by minimal residual disease measurements

PLOS ONE

Dear Dr. Martinez-Lopez,

Thank you for submitting your manuscript to PLOS ONE. After careful consideration, we feel that it has merit but does not fully meet PLOS ONE’s publication criteria as it currently stands. Therefore, we invite you to submit a revised version of the manuscript that addresses the points blow,  raised during the review process.

We look forward to receiving your revised manuscript.

Kind regards,

Nicola Amodio, PhD

Academic Editor

PLOS ONE

Journal Requirements:

2. Thank you for your ethics statement: 'The retrospective observational study was approved by both hospital ethics committees, and all patients had given their written informed consent to samples.'

(a) Please amend your current ethics statement to include the full name of the ethics committee/institutional review board(s) that approved your specific study.

(b) Once you have amended this/these statement(s) in the Methods section of the manuscript, please add the same text to the “Ethics Statement” field of the submission form (via “Edit Submission”).

3. We noticed minor instances of text overlap with the following previous publication(s), which need to be addressed:

(1) https://ashpublications.org/blood/article/126/7/858/34457/Critical-analysis-of-the-stringent-complete

The text that needs to be addressed involves the Introduction section.

In your revision please ensure you cite all your sources (including your own works), and quote or rephrase any duplicated text outside the methods section. Further consideration is dependent on these concerns being addressed.

4. We suggest you thoroughly copyedit your manuscript for language usage, spelling, and grammar. If you do not know anyone who can help you do this, you may wish to consider employing a professional scientific editing service.  

J Martinez-Lopez belongs to the speaker bureau of Adaptive Biotechnologies. The rest

of the authors declare no competing financial interests.

Reviewers' comments:

Reviewer's Responses to Questions

**Comments to the Author**

1. Is the manuscript technically sound, and do the data support the conclusions?

Reviewer #1: Partly

Reviewer #2: Yes

2. Has the statistical analysis been performed appropriately and rigorously? 

Reviewer #1: Yes

Reviewer #2: Yes

3. Have the authors made all data underlying the findings in their manuscript fully available?

Reviewer #1: Yes

Reviewer #2: Yes

4. Is the manuscript presented in an intelligible fashion and written in standard English?

Reviewer #1: Yes

Reviewer #2: Yes

5. Review Comments to the Author

Reviewer #1: Authors speculated about the role of MRD over sCR in the definition of response and that sCR should be replaced by MRD in th IMWG criteria. They demontsrated that in a coohrt of MM patiemts ther is no difference in outcome in pts < sCR vs > scR. Howvere some revisions could be taken into account:

- Abstract: in the conclusion maybe it would be more appropriate to say "MRD should be implemented over sCR.." since MRD is not yet in clinical practice and sCR could be still helpful nowdays.

- Methods: usually patients caractheristics are in the results part, I suggest to put the table and the explanation in the first part of results.

- MFC: probably MFC 4 color has too low sensitivity. How many events did you acquire? I think at least 8 colour MFC should be done ot at least to discuss a little bit in the discussion part the role of more sensitive techniques

Reviewer #2: The authors should comment more on how MRD evaluation should complement sCR more than replacing it. Moreover, 4-colors flow used in the study is not appropriate considering its low sensitivity over 8 or 10-colors flow. Authors should comment on that as well. It would be important to add patients characteristics in the results section. English should be checked.

6. PLOS authors have the option to publish the peer review history of their article (what does this mean?). If published, this will include your full peer review and any attached files.

Reviewer #1: No

Reviewer #2: No

---

## [Author Response · Author response to Decision Letter 0]

25 Jun 2020

June 25, 2020

Dear Editor/s of PlosOne,

First, we are grateful to you for considering and revising our work. 

Please find below our answer to your additional requirements.

Journal Requirements

The manuscript has been revised according to PLOS ONE’s style requirements. Particularly, affiliations and table 1 have been changed to suit your requirements. 

2. Thank you for your ethics statement: 'The retrospective observational study was approved by both hospital ethics committees, and all patients had given their written informed consent to samples.'

(a) Please amend your current ethics statement to include the full name of the ethics committee/institutional review board(s) that approved your specific study.

(b) Once you have amended this/these statement(s) in the Methods section of the manuscript, please add the same text to the “Ethics Statement” field of the submission form (via “Edit Submission”).

In the Methods section, we have included “The retrospective observational study was approved by Hospital 12 Octubre and the University of California San Francisco (UCSF) Ethics Committees”. 

Further, the same statement is included in the submission form.

3. We noticed minor instances of text overlap with the following previous publication(s), which need to be addressed:

(1) https://ashpublications.org/blood/article/126/7/858/34457/Critical-analysis-of-the-stringent-complete. 

The text that needs to be addressed involves the Introduction section. 

In your revision please ensure you cite all your sources (including your own works), and quote or rephrase any duplicated text outside the methods section. Further consideration is dependent on these concerns being addressed.

The text in the Introduction section have been modified to solve the overlap with our previous work. This publication had been cited, (Reference 8).

4. We suggest you thoroughly copyedit your manuscript for language usage, spelling, and grammar. If you do not know anyone who can help you do this, you may wish to consider employing a professional scientific editing service. 

Following to your suggestions, the editing service AJE has reviewed language usage, spelling and grammar. Editing revision is included as “Supporting information”.

J Martinez-Lopez belongs to the speaker bureau of Adaptive Biotechnologies. The rest

of the authors declare no competing financial interests.

A modified version of Competing Interests has been included in cover letter 

“Conflict of Interest: J Martinez-Lopez belongs to the speaker bureau of Adaptive Biotechnologies. This does not alter our adherence to PLOS ONE policies on sharing data and materials. The rest of the authors declare no competing financial interests.”

Review Comments to the Author

Reviewer #1: Authors speculated about the role of MRD over sCR in the definition of response and that sCR should be replaced by MRD in th IMWG criteria. They demontsrated that in a coohrt of MM patiemts ther is no difference in outcome in pts < sCR vs > scR. Howvere some revisions could be taken into account:

- Abstract: in the conclusion maybe it would be more appropriate to say "MRD should be implemented over sCR.." since MRD is not yet in clinical practice and sCR could be still helpful nowdays

We agree with your suggestion and have modified abstract conclusion as follows: 

“We suggest that MRD categories should be implemented over sCR for the future classification of MM responses”

- Methods: usually patients characteristics are in the results part, I suggest to put the table and the explanation in the first part of results.

Description of patients’ characteristics and the table have been moved to Results part.

- MFC: probably MFC 4 color has too low sensitivity. How many events did you acquire? I think at least 8 colour MFC should be done ot at least to discuss a little bit in the discussion part the role of more sensitive techniques

In the discussion section, we added that although our MFC is 4-color, the acquisition of at least 2 million of events increases the sensitivity above 10-4, much higher that achieved with immunochemistry (about 1%). More sensitive techniques to evaluate MRD, as next-generation flow or NGS, improve the predictions of outcome. This has been confirmed in a recently published study using next generation cytometry to study MRD. An 82% of reduction in the risk of death or relapse is showed in MRD negative patients.

Reviewer #2: The authors should comment more on how MRD evaluation should complement sCR more than replacing it. Moreover, 4-colors flow used in the study is not appropriate considering its low sensitivity over 8 or 10-colors flow. Authors should comment on that as well. It would be important to add patients characteristics in the results section. English should be checked.

4-color MFC has achieved an intermediate sensitivity if enough number of events are acquired. We include this explanation in discussion, as well as the relevance of more sensitive techniques for MRD analysis in the clinical setting. 

Patients characteristics have been moved to results section.

English has been checked by a native English spoken.

We hope we have answered appropriately your suggestions. 

Yours faithfully,

Joaquín Martínez-López

---

## [Decision Letter · Decision Letter 1]

22 Jul 2020

Clinical significance of stringent complete response in multiple myeloma is surpassed by minimal residual disease measurements

PONE-D-20-12394R1

Dear Dr. Martinez-Lopez,

We’re pleased to inform you that your manuscript has been judged scientifically suitable for publication and will be formally accepted for publication once it meets all outstanding technical requirements.

Kind regards,

Nicola Amodio, PhD

Academic Editor

PLOS ONE

Additional Editor Comments (optional):

Reviewers' comments:

Reviewer's Responses to Questions

**Comments to the Author**

1. If the authors have adequately addressed your comments raised in a previous round of review and you feel that this manuscript is now acceptable for publication, you may indicate that here to bypass the “Comments to the Author” section, enter your conflict of interest statement in the “Confidential to Editor” section, and submit your "Accept" recommendation.

Reviewer #1: All comments have been addressed

Reviewer #2: All comments have been addressed

2. Is the manuscript technically sound, and do the data support the conclusions?

Reviewer #1: Yes

Reviewer #2: Yes

3. Has the statistical analysis been performed appropriately and rigorously? 

Reviewer #1: Yes

Reviewer #2: N/A

4. Have the authors made all data underlying the findings in their manuscript fully available?

Reviewer #1: Yes

Reviewer #2: Yes

5. Is the manuscript presented in an intelligible fashion and written in standard English?

Reviewer #1: Yes

Reviewer #2: Yes

6. Review Comments to the Author

Reviewer #1: (No Response)

Reviewer #2: The authors have addressed all the concerns by this reviewer. The paper can now be accepted for publication.

7. PLOS authors have the option to publish the peer review history of their article (what does this mean?). If published, this will include your full peer review and any attached files.

Reviewer #1: No

Reviewer #2: No

---

## [Editor Report · Acceptance letter]

24 Jul 2020

PONE-D-20-12394R1 

The clinical significance of stringent complete response in multiple myeloma is surpassed by minimal residual disease measurements 

Dear Dr. Martinez-Lopez:

I'm pleased to inform you that your manuscript has been deemed suitable for publication in PLOS ONE. Congratulations! Your manuscript is now with our production department. 

Kind regards, 

on behalf of

Dr. Nicola Amodio 

Academic Editor

PLOS ONE